# Development and Characterization of the Biodegradable Film Derived from Eggshell and Cornstarch

**DOI:** 10.3390/jfb13020067

**Published:** 2022-05-27

**Authors:** Joseph Merillyn Vonnie, Kobun Rovina, Rasnarisa Awatif Azhar, Nurul Huda, Kana Husna Erna, Wen Xia Ling Felicia, Md Nasir Nur’Aqilah, Nur Fatihah Abdul Halid

**Affiliations:** 1Faculty of Food Science and Nutrition, Universiti Malaysia Sabah, Kota Kinabalu 88400, Sabah, Malaysia; vonnie.merillyn@gmail.com (J.M.V.); rasnarisa@gmail.com (R.A.A.); drnurulhuda@ums.edu.my (N.H.); mn1911017t@student.ums.edu.my (K.H.E.); felicialingling.97@gmail.com (W.X.L.F.); aqilah98nash@gmail.com (M.N.N.); 2Borneo Marine Research Institute, Universiti Malaysia Sabah, Kota Kinabalu 88400, Sabah, Malaysia; fatihahhalid@ums.edu.my

**Keywords:** biodegradable film, starch, calcium carbonate, eggshell, physicochemical

## Abstract

In the current study, cornstarch (CS) and eggshell powder (ESP) were combined using a casting technique to develop a biodegradable film that was further morphologically and physicochemically characterized using standard methods. Scanning electron microscopy (SEM) and transmission electron microscopy (TEM) were used to characterize the morphology of the ESP/CS film, and the surface of the film was found to have a smooth structure with no cracks, a spherical and porous irregular shape, and visible phase separation, which explains their large surface area. In addition, the energy dispersive X-ray (EDX) analysis indicated that the ESP particles were made of calcium carbonate and the ESP contained carbon in the graphite form. Fourier Transform Infrared Spectroscopy indicated the presence of carbonated minerals in the ESP/CS film which shows that ESP/CS film might serve as a promising adsorbent. Due to the inductive effect of the O–C–O bond on calcium carbonate in the eggshell, it was discovered that the ESP/CS film significantly improves physical properties, moisture content, swelling power, water solubility, and water absorption compared to the control CS film. The enhancement of the physicochemical properties of the ESP/CS film was principally due to the intra and intermolecular interactions between ESP and CS molecules. As a result, this film can potentially be used as a synergistic adsorbent for various target analytes.

## 1. Introduction

Eggshell (ES) is a byproduct of poultry production and kitchen waste. Most country to dispose of these waste products primarily in landfills without additional treatment, which has serious environmental consequences [1]. Malaysia produces approximately 642,600 tons of eggs per year and is expected to generate 70,686 tons of ES waste per year [2]. This is due to the fact that eggs are a common ingredient in a variety of products, including cakes, fast food, and everyday meals. However, statistics show that egg production in Malaysia has increased dramatically to 773 thousand tonnes, with an expected annual growth rate of 3.6% by 2020. In this case, ES waste can be used as a source of calcium carbonate (CaCO_3_), which accounts for approximately 95% of a shell mass and 5% of organic materials such as sulfated polysaccharides, collagen, and other proteins [3]. Furthermore, CaCO_3_ has been used to remove manganese, cadmium, lead, zinc, nickel, and chromium [4]; additionally, CaCO_3_ is known for having good morphology and porosity and removing more than 90% of metal ions when the pH, contact time, agitation speed, and adsorbent dosage are optimized. Previously, Cree and Rutter [5], as well as Betancourt and Cree [6], have conducted morphological studies to compare mineral limestone and eggshell powder (ESP) under a scanning electron microscope at the same magnification. ESP revealed the presence of pores, while these were absent in the limestone. This implies that eggshells have bigger pores than limestone, which allows them to serve as an adsorbent.

One of the greatest threats to human life is environmental pollution caused by synthetic polymers, plastics, and non-degradable packaging materials entering the environment [7]. Thin film-based biopolymers, such as starch, have become increasingly important in the manufacturing sector, particularly in the food industry. Biodegradable films are typically composed of polypeptides, lipids, polysaccharides, or a combination of these materials. The development of biodegradable films made from renewable natural resources has been investigated as a good alternative to reduce the negative environmental impacts caused by waste disposal as they can easily be decomposed by the environment [8,9]. Corn starch is a natural biopolymer that has been widely employed in the development of environmentally friendly packaging materials [10]. Moreover, corn is very rich in starch, which is one of the foremost copious and biodegradable biopolymers and is commonly utilized within the food, pharmaceutical, textile, biomass energy, and chemical industries [11]. However, due to its continuous matrix formation capacity and high amylose content, starch has a higher film-forming capability when compared with other polysaccharides [12,13]. Starch has numerous benefits in the development of biodegradable films, including low cost, renewability, appropriate physical, chemical, and functional properties, and a diverse source base [14,15]. Furthermore, starch is a natural carbohydrate polymer derived from plants, such as cereals, roots, and tubers. Several novel bio-based materials have been investigated as a potential alternative to resolve environmental issues, including wheat [16], corn [17], potato [18], cassava [19], sugar palm starch [20], and pea [21]. ESP/CS film was employed as a potential polymeric adsorbent for the adsorption and removal of analytes.

The previous research indicates that ES has the potential to be a good sorbent due to its high carbon and calcium concentrations, as well as its high porosity and good accessibility of functional groups. Recently, Wang et al. [22] studied the application of granular bentonite−eggshell composites (BEP) for the removal of heavy metal (Pb). BEP had an elimination ratio of over 99.90% and an adsorption capacity of over 40 mg/g of Pb, respectively. The Elovich kinetic model described the Pb adsorption process well and suggested removal processes, including ion exchange, electrostatic adsorption, and complexation, between BEP and Pb. Isa et al. [23] proposed the use of two natural agro-wastes composed of eggshells and sugarcane bagasse as potential adsorbents in the removal of Pb (II) and Cd (II) from aqueous solutions. The maximum concentrations of Pb and Cd absorbed per gram of biosorbent were determined to be 277.8 and 13.62 mg/g for eggshells and 31.45 and 19.49 mg/g for bagasse, respectively. Antecedently, Mostafavi et al. [24] designed and fabricated nanocomposite-based polyurethane filters with excellent physical and chemical adsorption capabilities for particular contaminants and the ability to alleviate or minimize pollutants from wastewater. This filter could be a promising device for the removal of heavy metals from water.

In this study, there is a focus on developing and characterizing the eggshell powder for the development of biodegradable films containing cornstarch since it is abundant, cost-effective, easily processed, and demonstrates strong potential as an adsorbent. The surface morphology, functional groups, and hydrophobic characteristics of composite film were studied. This research will serve to remediate and alleviate the side effects of contaminants in various types of food and wastewater in the industrial sectors.

## 2. Materials and Methods

### 2.1. Materials

Raw and fresh eggshells were collected from a local restaurant in Sabah, Malaysia. Cornstarch powder (purity 100%) and potassium bromide (KBr, purity ≥ 99%) were purchased from Sigma Aldrich, St. Louis, MO, USA.

### 2.2. Preparation of Eggshell Powder (ESP)

Fresh ES was collected and rinsed several times with standard tap water followed by deionized water to eliminate contaminants and interfering materials on the surface of the ES. Then, the washed ES was air dried for 30 min and dried for 5 h at 50 °C in a hot air oven (Binder, 07-32195). The dried ES was then blended into fine particles using a blender (Panasonic MX-337) and sieved through a 250 m sieve to obtain the finely powdered sample, which was then stored in an airtight container [25,26].

### 2.3. Preparation of Biodegradable ESP/CS Films

ES solution was prepared by grinding 0.6 g of ESP and dispersing it in 30 mL of distilled water. The mixtures were continuously stirred at room temperature for 2 h to thoroughly wet the ESP particles before being filtered through Whatman filter paper. Following that, 0.6 g of CS powder was dissolved in 10 mL of distilled water and ES solution to produce control films of CS film solution and ESP/CS film solution, respectively. The solutions were then stirred with a magnetic stirrer at 100 °C until gelatinized. Each film-forming solution was poured into 90 mm internal diameter Petri dishes and allowed to form films at room temperature for nearly 24 h.

### 2.4. Morphological Characterization

The surface morphology and elemental composition of ESP/CS and CS films were analyzed using scanning electron microscopy (SEM, Carl Zeiss Ma 10), energy-dispersive X-ray spectroscopy (EDX), and transmission electron microscopy (TEM, Tecnai G Spirit Biotwin).

### 2.5. Physical Characterization

#### 2.5.1. Thickness (*e*) and Density (*ρ*)

A micrometer with a precision of ±0.001 mm was used to measure the thickness of developed films. The film dimensions were determined by taking samples at five different locations [27]. For the *ρ* determination, 6.25 cm^2^ (area) sample discs from each film were used. The *ρ* film was then calculated as the ratio of weight (*W*) to volume (*V*), where *V* equals area (*A*), by multiplying the *e* of each film as in Equation (1).
(1)$$\rho =\frac{w}{v}=\frac{w}{A*e\;}$$

#### 2.5.2. Moisture Content (*MC*), Water Solubility (*WS*), Water Absorption (*WA*) and Swelling Power (*SP*) of Films

For *MC*, the films were cut into 2.5 cm × 2.5 cm pieces and weighed (*W_i_*). Then, it was dried at 105 °C for 24 h and weighed as *W_f_*. The MC of the different films was calculated using Equation (2) [28].
(2)$$MC\;\left(\%\right)=\frac{W_{i}-W_{f}\;}{W_{i}}\times 100$$

For *WS*, the initial dry weight (*W_i_*) of film was determined by drying it at 105 °C for 24 h. Each sample was then immersed in 50 mL of H_2_O and kept at 25 °C for 24 h. Subsequently, the films were dried in an oven at 105 °C for 24 h and weighed (*W_f_*). The data obtained was used to calculate the *WS* for each film using Equation (3).
(3)$$WS\;\left(\%\right)=\frac{W_{i\;}-W_{f\;}}{W_{i}}\times 100$$

The *WA* of the films was determined according to Wu et al. [25]. The films (2.5 cm × 2.5 cm) dried at 105 °C for 24 h, and the initial weight (*W_i_*) was obtained. The films were then immersed in H_2_O at room temperature for 60 min. The samples were taken out and weighed immediately after wiping off the H_2_O surface with filter papers (*W_f_*). The percentages of *WA* were calculated using Equation (4).
(4)$$WA\;\left(\%\right)=\frac{W_{f}-W_{i}\;}{W_{i}}\times 100$$

The *SP* % of the developed films was determined according to the method described by Herniou et al. [29]. Each of the film samples was initially cut into 2 cm diameter (ø*_i_*) discs and immersed in 20 mL of H_2_O. The containers were then sealed and maintained at room temperature for 24 h. The disc diameters (ø*_f_*) of the samples were recorded and calculated using Equation (5).
(5)$$SP\;\left(\%\right)=\frac{\emptyset_{f}-\emptyset_{i}}{\emptyset_{i}}\times 100$$

### 2.6. Storage Stability

The storage stability of the films was determined in three different conditions by measuring the weight loss (*W*_L_) over 28 days. Three containers consisting of five replicate film samples were stored in the incubator (37 °C), cold room (4 °C), and room temperature. The weights were recorded on day 0, 7, 14, 21, and 28. The W_L_ was calculated using Equation (6).
Weight Loss (*W*_L_) = *W_i_* − *W_f_*
(6)

### 2.7. Fourier Transform Infrared (FTIR) Spectroscopy and X-ray Diffraction (XRD) Analysis

CS and ESP/CS films’ functional groups were measured using an FTIR spectrometer (Perkin-Elmer Universal Attenuated Total Reflectance spectrometer). The film samples were blended with potassium bromide (KBr) powder and pressed into a pellet. The crystalline structure and particle size of the film samples were observed by X-ray diffraction (XRD) analysis. The XRD patterns were recorded using a diffractometer (PANalytical-Empyrean instrument; Co radiation: 1.54056 A°) and analyzed between 0 and 90° (2 theta). The voltage, current, and amount of time passed were 40 Kv, 40 mA, and 1 s.

### 2.8. Statistical Analysis

The results are expressed as the mean ± standard deviation (n = 5) to ensure the data’s accuracy. The SPSS IBM Statistics 19 software was used to analyze the data. The experimental data were analyzed using one-way ANOVA, with Tukey Post-Hoc test at a 95% confidence interval with *p* < 0.05 to establish significant differences between the samples.

## 3. Results and Discussion

### 3.1. Morphology Characterization

Figure 1 shows the surface of the CS and ESP/CS films, as well as their composition, which was determined by scanning electron microscopy (SEM). CaCO_3_ is the primary component of ES and is composed of three polymorphs: vaterite, aragonite, and calcite. These polymorphs are temperature-dependent, with different temperatures producing different CaCO_3_ structures with different morphology [30]. Figure 1a,b shows the spheroidal-shaped crystallites of vaterite, as well as the smallest crystallites, which are about 1–2 µm in size [31]. The porous surface textures of the ESP/CS films support the adsorbent by increasing the surface area and allowing the adsorption process to occur while also exhibiting a non-adhesive appearance and agglomeration formation. Due to its high adsorption capacity, raw ES’s porous external and internal layers made it an appealing bioadsorbent. Furthermore, the external surface of the ES is primarily composed of CaCO_3_, whereas the inner layer is made up of uncalcified fibrous membranes composed primarily of organic compounds [32].

The surfaces of the ESP/CS films, as shown in Figure 1b, have a homogeneous and smooth structure with no cracks, visible phase separation, and embedded ESP in the starch matrix. According to Sun et al. [33], phase separation could destabilize the interface adhesion between the nanofiller and the matrix, lowering the tensile strength of films with increased nanoparticle content. As a result of the organic components in ESP, the ESP particles are spread uniformly in the film matrix, resulting in improved adhesion to the starch matrix [34]. The microstructure of the ESP reveals that the particle size and shape varies; however, they are composed of porous irregularly-shaped particles. According to King’Ori [35], the microstructure reveals relatively uniform distributions of ESP particles in the matrix. The distribution of ESP particles is influenced by the ESP particles’ excellent wettability by molten metal and good interfacial bonding between the particles and matrix material. In the microstructures of the composites, the ESP was well retained and distributed along the grain boundaries.

The granular shape of CS was round and polygonal, as shown in Figure 1c,d, but the surface became rough as many pores formed on the starch granules, resulting in a compact structure [34]. Some of the pores even extend from the surface to the interior of the starch granules, as shown in Figure 1d, and the starch granules have large interval cavities. Due to these structural properties, a large specific surface area is possible. As a result, it is possible to conclude that these macropores can improve the adsorption of adsorbates and can be used as adsorbents in a variety of applications [36,37]. Furthermore, starch granules with various sizes or apparent densities had inherent crystallinity variations, which influenced the granule’s functional properties.

The EDX of the ESP particles demonstrates that the particles contain calcium (Ca), oxygen (O), copper (Cu), gold (Au), nitrogen (N), and silica (Si), with a presence of carbon (C) (61.83%) in the ESP particles. According to King’Ori [35], ES contains approximately 98.2% of calcium carbonate, 0.9% of magnesium, and 0.9% of phosphorus, which showed that the ESP particles are made of CaCO_3_ and contain carbon in the graphite. These analyses are comparable to other authors’ reinforcement analyses. The microstructure of the unreinforced Au (16.42%) and Cu (1.87%) also contained in this sample led to the preparation of a sample in which all specimens (external surfaces) were coated with a thin layer of gold under a high vacuum. According to the EDX analysis of the composite material, there is a possibility of a chemical reaction between aluminium (Al) melt and the ESP particle, which resulted in the release of Si (0.57%), N (11.94%), O (19.77%), C, and Ca (0.12%) [38].

Next, TEM was used to determine the characteristics of the various materials, including the morphology, shape, and particle size of ESP/CS and CS. The TEM images in Figure 1e,f show that the particles have irregular shapes, which explains their large surface area [39]. In addition, irregular shapes in ES have been observed, which may be due to the calcite shape of the ESP [40]. According to Minakshi et al. [39], bright-field TEM imaging of the ES revealed dense and angular calcite crystals with particle sizes ranging from 30 to 500 nm. The bright field can be seen mostly in Figure 1f and in some parts of Figure 1e. The porous and nanocrystalline nature can be seen in the high-resolution TEM image (Figure 1g). This occurred because the nucleation and growth of CaO crystals from the CaCO_3_ parent contributed to a roughened surface texture on the particles [39].

In contrast, high-resolution TEM measurements revealed that the CS particles were small, had good uniformity, and an almost perfect spherical shape, with a diameter ranging from 53.1 nm–83.1 nm (Figure 1h). The surface of the CS film without any ESP was smooth. As shown in Figure 1h, the spherical particles and the particle size distribution were relatively homogeneous and no agglomeration occurred. The particles of CS in Figure 1i consist of both large and small irregular bodies entangled by thin filaments which have no encapsulated structures, and more dispersion could be observed. However, Ponsanti et al. [41] intensely observed that CS has more dispersion and a smaller particle size than other starches, including cassava starch and sago starch.

### 3.2. Film Thickness and Density Measurement

After the addition of ESP, the thickness of the composite film changed. The thickness of the CS film was 0.018 ± 0.021 mm, whereas the thickness of the ESP/CS film was increased to 0.026 ± 0.018 mm due to the presence of CaCO_3_ particles in the ESP, which strengthened the interfacial interaction between ESP and the CS film matrix, hence reducing matrix chain mobility and increasing the macroscopic thickness of the ESP/CS composite film. At the same time, incorporation of ESP significantly increased the density of ESP/CS film due to the establishment of a continuous network facilitated by the incorporation of ESP, which minimized the effect of water [34].

### 3.3. Moisture Content (MC)

The CS film had a higher *MC* value (48.533 ± 18.213) than the ESP/CS film (38.781 ± 16.139), which was due to the greater number of available OH groups in the starch film. Furthermore, this led to the outstanding ability of CS film to absorb moisture from the environment, and this finding was compatible with the FTIR spectra analysis. The lower *MC* value of the ESP/CS film was due to the cross-linking between ES (CaCO_3_) and CS, which reduced the polarity and the interaction between hydrophilic groups and water [42]. The objective of determining the film’s moisture content was to ascertain its superior removal capacity. The moisture content of a film indicates the percentage of water capacity present. The lower the moisture content, the greater the adsorption efficiency due to lesser water molecules binding to the active sites of the film matrix and allowing target analytes to bind onto the active sites [43]. Therefore, the ESP/CS film showed promise as an adsorbent due to its lower moisture content.

### 3.4. Water Solubility (WS)

The ESP/CS film (39.022 ± 12.251) displayed a lower solubility than the CS film (46.632 ± 9.109), as shown in Table 1. The intramolecular interaction and arrangement of the ESP and CS granules can be attributed to these outcomes. CS in ESP solution can form highly cross-linked systems, preventing water molecules from penetrating the ESP/CS films and dissolving ESP fiber proteins and starch granules [13]. Consequently, the capacity of the film to absorb water was reduced [44] due to the greater amount of ESP filler loaded onto the film matrix [45]. The ESP/CS film showed a moderate decrease in WS during the gelatinization process due to the interaction with ES molecules where the degradation of ES and starch co-occurred, wherein the hydroxyl groups (-OH) of degraded starch interacted with the functional groups of ES, resulting in the formation of a cross-linking network [13]. The reduced number of free -OH groups made the ESP/CS film less attractive to water molecules, which also helped to slow down the release rate of free polymer chains from collagen and starch to water, and eventually resulted in the lower solubility of ESP/CS films compared to CS films [13,46].

### 3.5. Water Adsorption (WA)

The presence of free hydroxyl groups (–OH) in ESP/CS film can result in high sorption of target analytes. Table 1 shows the *WA* value for ESP/CS and CS films. It showed that both films absorbed water very well during the 60 min immersion process to reach the saturation level. The addition of ESP into CS can increase the *WA* of ESP/CS film (87.700 ± 3.374) compared to CS film (87.072 ± 2.758). These results indicated that a higher content of filler and the presence of high levels of CaCO_3_ in the ESP, thus increasing the *WA* of ESP/CS film. In addition, the high *WA* value in the ESP/CS film with higher ESP loading was due to the increased number of voids between ESP and CS matrices [47].

### 3.6. Swelling Power (SP)

Based on Table 1, the *SP* value of the ESP/CS film was reduced compared to the CS film with values of 7.000 ± 2.739 and 8.000 ± 2.739, respectively. A decreasing *SP* value of the ESP/CS film indicated better resistance to swelling, probably due to the presence of more vital associative forces maintaining the granule structure [48]. However, the rise in the *SP* of the CS films is attributed to the hydrogen bond in water molecules to the exposed hydroxyl groups of amylose and amylopectin.

### 3.7. Storage Stability

The storage stability test applied to the films aimed to provide evidence on how the quality of the films varies with time under the influence of environmental factors, including temperature, humidity, and light. The storage stability for both films was analyzed for 28 days at three different places with different temperatures, as shown in Figure 2. The storage time and temperature affected water activity. From the obtained result, the trends for both biodegradable films started to decrease after day 7, which was attributed to the decreasing water-holding capacity of the films and the moisture loss [49].

Based on Figure 2, the storage stability for ESP/CS film kept at room temperature (25 °C) and in an incubator (37 °C) was higher relative to those kept in a cool room (4 °C). Previously, Jiang et al. [34] reported that composite films consisting of CS and ESP have excellent thermal stability due to the presence of ESP which affects the matrix’s heat conductivity and prevents the starch films from degrading thermally. Moreover, it was reported that CS and ESP show good compatibility. For example, ESP was tightly dispersed in the matrix, resulting in increased thermal stability and an extended degradation rate of the composite [50]. However, the storage stability of the ESP/CS film was lower in refrigerated conditions due to the slower active reaction of the composite film [51]. Also, the ESP/CS film provided a significantly higher quality of film at room temperature when compared with other conditions; the ESP/CS film was 0.9370 g, while the CS film was 0.9011 g after 28 days. This scenario happened because ESP/CS film had a lower peroxide value and a higher induction period than CS film, indicating greater stability [52]. The peroxide values increased faster at room temperature than at lower temperatures (cool room) during the storage time, and the highest storage stability for the ESP/CS film stored at room temperature was exposure to daylight. Thus, higher peroxide values and a longer induction period reduced the decomposition of the ESP/CS film and provided greater storage stability.

### 3.8. Fourier Transform Infrared (FTIR) and X-ray Diffraction (XRD) Analysis

The results of the stretching and bending vibrations of the functional groups in the ESP/CS, which is responsible for the adsorption of the adsorbate molecules, are shown in Figure 3. Figure 3a shows the IR spectral analysis for ESP/CS with a distinct peak at 654.34 cm^−1^, 712.13 cm^−1^, 872.42 cm^−1^, and 1400.03 cm^−1^. An intense peak and a strong absorption band of the ES particle was observed at 1400.03 cm^−1^, which can be strongly associated with the presence of carbonate minerals within the ES matrix [53]. There are also two observable sharp bands at 712.13 and 872.42 cm^−1^, which were associated with the asymmetric stretching of the C-H band assigned and the in-plane deformation and out-plane deformation, indicating the existence of CaCO_3_ [53]. Equally significant, the prominent absorption peaks of carbonates at 1400.03 cm^−1^ support Onwubu et al.’s [1] findings. They claimed that carbonate-based materials commonly detected the broad stretching frequency of the C=O bond in carbonate ions.

Figure 3b shows the characteristic peaks of CS at 3269.84 cm^−1^ (O–H stretching), 2930.55 cm^−1^ (CH_2_ group), 1636.78 cm^−1^ (CH–O–H group), and 1075.65 cm^−1^ (C–O stretching) [53]. The absorption band occurred in CS at approximately 652.48 cm^−1^, reflecting the typical stretching vibrations of C-C skeletons. The peak and strong absorption band at 3269.84 cm^−1^ were attributed to the presence of functional groups containing O-H and N-H stretching of starch and carboxylic acid (-COOH) groups at 1636.78 cm^−1^ and 1075.65 cm^−1^ wavelengths, respectively, which has a significant effect on the process of adsorption and the efficiency of adsorbing target analytes via ion exchange [54,55]. Moreover, a strong band observed at 1636.78 cm^−1^ was attributed to the water adsorbed in the amorphous amylose region of starch [56]. These results indicated an increase in the number of hydrogen bonds between the starch molecules and the ESP particles, which led to the vibrational frequency of the pyranoid ring skeleton decreasing and the absorption bands shifting to a lower wavenumber. It showed that the addition of ESP promoted the formation of hydrogen bonds and reduced the number of active sites for water adsorption, which decreased the free space in the film network and strengthened the compact structure of composite films [57]. Thus, the water and oxygen resistance abilities of the ESP/CS films improved.

The crystalline structure of starch granules consists of the ordered crystalline region and the disordered amorphous region. X-ray diffraction (XRD) patterns were used to evaluate the amorphous crystalline structure characterized by sharp peaks related to crystalline diffraction and an amorphous zone [58]. The wide-angle XRD patterns of the ESP/CS and CS films with the presence and absence of the particles in between 10°(2θ) and 50°(2θ) are shown in Figure 4. A new strong characteristic reflection appearing at 2θ = 29.4° can be seen from the XRD patterns of the ESP/CS films, which led to the semi-crystalline structure of the composite films. In addition, the greater intensity of the diffraction peaks at 2θ = 29.4° may be related to the amount of ESP, attributable to the calcite crystal and agglomerates of ESP. Also, the peak broadening is caused by small crystal size. This indicated that the ESP was uniformly dispersed in the CS matrix and built a strong interaction with CS [34]. However, the XRD pattern shown in Figure 4 was similar to the commercial calcium carbonate pattern exhibited by previous research by Ji et al. [59], where it showed that the main component of eggshell powder was 95% CaCO_3_. The peak of the ESP/CS film at 8.5° disappeared in the nanocomposite films, which led to the excellent compatibility between the CS and CaCO_3_ particles. As expected, they modified the peak intensity of the films, where the ESP/CS film produced slightly higher peak intensities than the CS film, resulting in the greater strength of CS films.

## 4. Conclusions

In this research, the ESP/CS film was prepared by a solution-casting method. The ESP particles were uniformly dispersed and embedded in the film matrix according to SEM analysis. In addition, SEM analysis of ESP/CS films showed a homogeneous and smooth structure with no cracks and visible phase separation. The image from TEM revealed irregular shapes of particles, which explains their larger surface area. The integration of ESP enhanced the thickness, density, water absorption, and swelling power in the film when compared to CS film. Moisture content, water solubility, and swelling power of ESP is lower than CS. The storage stability of ESP is higher in room temperature and in an incubator. Furthermore, FTIR analysis revealed that the addition of ESP aided the formation of hydrogen bonds and strengthened the compact structure of composite films, as well as the potential to produce a strong absorption band, which can be strongly associated with the presence of carbonate minerals. XRD analysis showed the semi-crystalline structure of the composite film and that the peak intensity of the ESP/CS film is higher than the CS film, which proves that ESP built strong interactions with CS. Finally, as a biofiller, ESP can improve the mechanical strength, hardness, and barrier properties of films without using cross-linking agents. This ESP/CS film is suitable to be used for the adsorption of target analytes, such as contaminants that are present in food products.

## Figures and Tables

**Figure 1 jfb-13-00067-f001:**
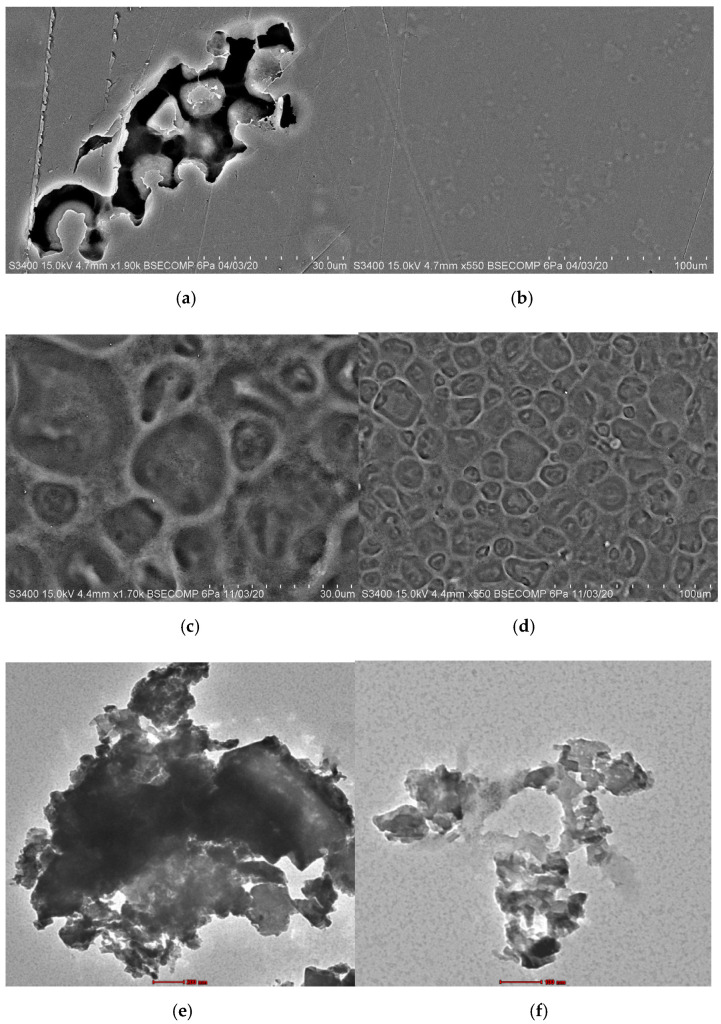
The SEM images of biodegradable (**a**,**b**) ESP/CS and (**c**,**d**) CS film with different magnifications (×30 µm, ×100 µm) and TEM images of (**e**–**g**) ESP/CS and (**h**,**i**) CS film with different magnifications (×100 µm, ×200 µm, ×500 µm).

**Figure 2 jfb-13-00067-f002:**
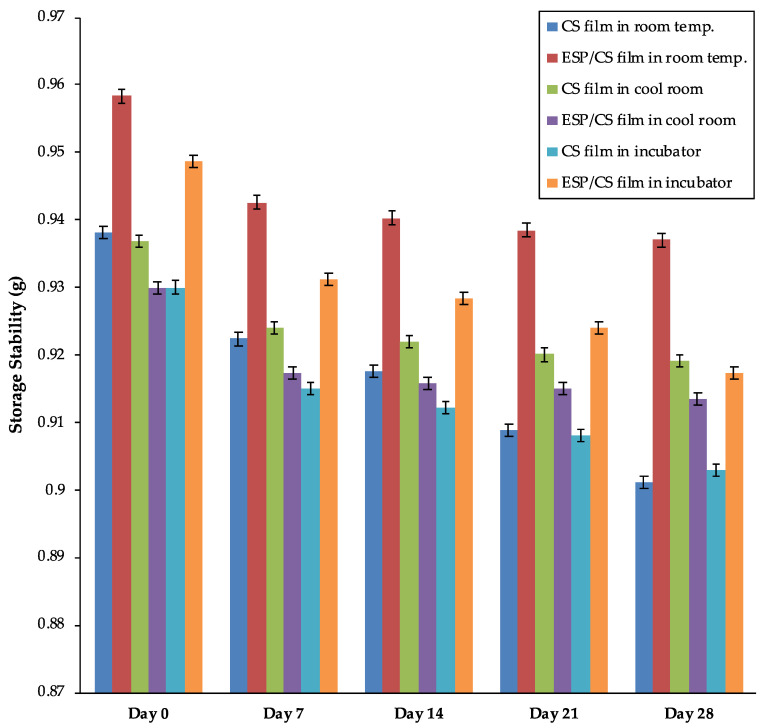
The storage stability of films for day 0, 7, 14, 21, and 28.

**Figure 3 jfb-13-00067-f003:**
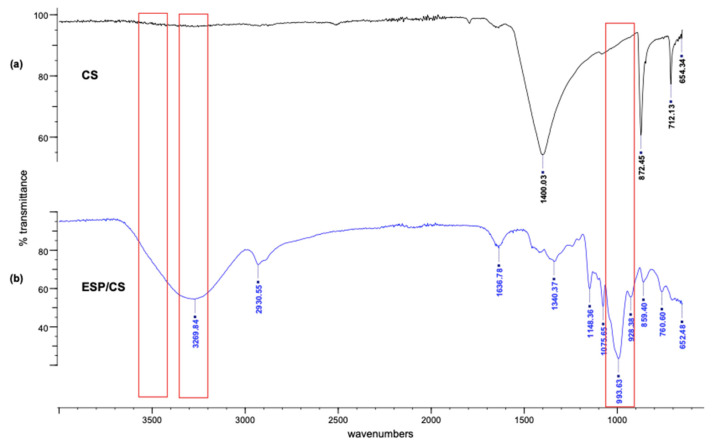
FTIR spectra of (**a**) ESP and (**b**) CS films.

**Figure 4 jfb-13-00067-f004:**
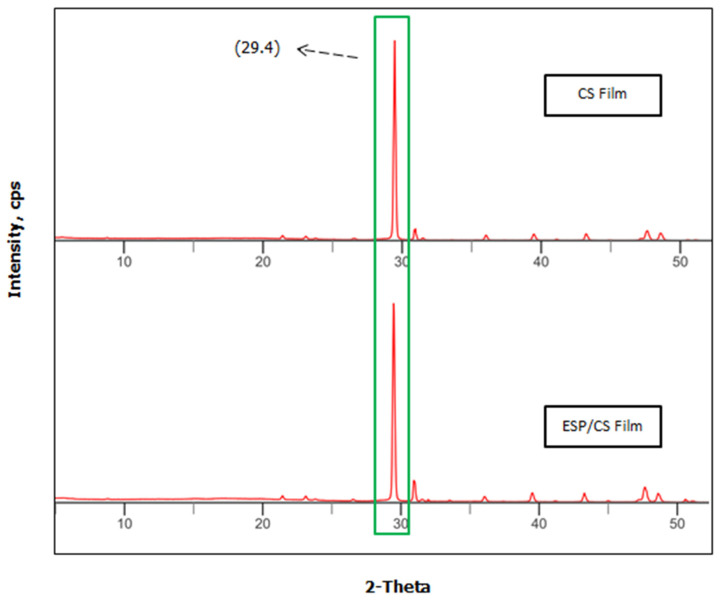
The wide-angle XRD patterns of the CS and ESP/CS films.

**Table 1 jfb-13-00067-t001:** The physicochemical values of CS and ESP/CS film.

Parameters	CS Film	ESP/CS Film
Thickness	0.018 ± 0.021 ^a^	0.026 ± 0.018 ^b^
Density	0.535 ± 0.386 ^c^	0.617 ± 0.210 ^d^
*MC* (%)	48.533 ± 18.213	38.781 ± 16.139 ^e^
*WS* (%)	46.632 ± 9.109	39.022 ± 12.251
*WA* (%)	87.072 ± 2.758	87.700 ± 3.374
*SP* (%)	8.000 ± 2.739	7.000 ± 2.739

Different lowercase letters in the same column indicate a statistically significant difference (*p* < 0.05).

## Data Availability

Not applicable.

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
