# Peer review of "Development and Characterization of the Biodegradable Film Derived from Eggshell and Cornstarch"

_jfb, 2022, doi:10.3390/jfb13020067_

Round 1

Reviewer 1 Report

Joseph et all described the preparation and physicochemical characterization of eggshell powder/Corn starch film for adsorption of heavy metals.

The overall merit of the article is suitable for publication but would like clarity/explanation on the following.

The stability of the films was characterized based on the water holding capacity of the films, the bar graph shows no error bars indicating at only one replicate is used to assess this. Indicate the number of replicates used for each assessment.

Also, the heavy metal adsorption capacity of this biofilm should be compared to other conventional techniques. Briefly describe the detection limits, and efficiency of the current films.

Author Response

Reviewer 1

Joseph et all described the preparation and physicochemical characterization of eggshell powder/Corn starch film for adsorption of heavy metals. The overall merit of the article is suitable for publication but would like clarity/explanation on the following.

  1. The stability of the films was characterized based on the water holding capacity of the films, the bar graph shows no error bars indicating at only one replicate is used to assess this. Indicate the number of replicates used for each assessment. 
  • Figure 2 (stability) has been updated based on the comments. The error bars can be found in the figure.

  1. Also, the heavy metal adsorption capacity of this biofilm should be compared to other conventional techniques. Briefly describe the detection limits, and efficiency of the current films. 
  • The manuscript is focusing on the development and characterization of the eggshell powder/Corn starch film which could be applied in the adsorption of heavy metals. Thus, the title has been revised into “Development and characterization of the biodegradable film derived from eggshell and cornstarch”

Reviewer 2 Report

Cornstarch and eggshell powder were combined by casting technique. films prepared bu these composing powder should strong heavy metal removing ability in food products. It is interesting and applicable. But some important data should be added. 
1. The degradation process of composites should be supported by corresponding experimental data or literature.
2. Thermodynamic characteristics should be supplemented, including glass transition temperature, thermal decomposition temperature, thermal stress parameters of composites and so on. 

Author Response

Reviewer 2

Cornstarch and eggshell powder were combined by casting technique. Films prepared by these composing powder should have strong heavy metal removing ability in food products. It is interesting and applicable. But some important data should be added. 

  1. The degradation process of composites should be supported by corresponding experimental data or literature.
  • Thank you for the comment. Corresponding experimental data or literature have been added under section 4.7 Storage stability in paragraph 2.

  1. Thermodynamic characteristics should be supplemented, including glass transition temperature, thermal decomposition temperature, thermal stress parameters of composites and so on. 
  • Thank you for the comments and suggestions. The authors totally agree with the reviewer, however, the current manuscript is focusing on the physical characterization of the film and we hope could be applied as an indicator in the presence of heavy metals. 

Reviewer 3 Report

The authors present their work on composite films made with eggshells (ES) and corn starch (CS) to be used for biodegradable food storage materials with additional feature for heavy metal removal.  The water interaction properties and storage stability of CS films were compared to the new ES/CS composites where the ES adds calcium carbonate aggregates to the starch matrix.

The approach to evaluating the films' properties is sound and identifies relevant considerations for the proposed applications.  However, the data seems to show that there is no difference in many, if not all, of the properties.  The authors mention statistical tests but provide no outcomes of those tests.  At first glance, it appears that there is no statistical difference between the CS and ES/CS films.  The discussion and conclusions mention that the ES/CS outperforms the CS film when that does not appear to be the case. In addition, the justification for how the films interact with water seems to be contradictory for different measurements (density compared to moisture content, for instance). 

Other notes - Text says that Figure 3a is ES/CS composite, but figure seems to show ES only.  Would be helpful to have all three (ES, CS, and ES/CS) for the FTIR. Figure 4 seems to show the same XRD pattern even though it should be CS and ES/CS comparison.  Don't know if these two were mistakes.  Error bars on the Storage Stability graph would be helpful.  Additionally, it doesn't look like there is data for ESP/CS film at room temp.

Because of these concerns, the manuscript is not recommended for publication.

Author Response

Reviewer 3

The authors present their work on composite films made with eggshells (ES) and cornstarch (CS) to be used for biodegradable food storage materials with an additional feature for heavy metal removal. The water interaction properties and storage stability of CS films were compared to the new ES/CS composites where the ES adds calcium carbonate aggregates to the starch matrix. The approach to evaluating the films' properties is sound and identifies relevant considerations for the proposed applications. 

  1. However, the data seems to show that there is no difference in many, if not all, of the properties. The authors mention statistical tests but provide no outcomes of those tests.  At first glance, it appears that there is no statistical difference between the CS and ES/CS films.  
  • Thank you for the comments. The statistics data has been updated according to the suggestion is shown in Table 1.

  1. The discussion and conclusions mention that the ES/CS outperforms the CS film when that does not appear to be the case.
  • The discussion and conclusion sections have been revised accordingly to show that ESP/CS outperformed CS film. Besides, statistical analysis has been conducted to show the superior properties of ESP/CS than CS film. 

  1. In addition, the justification for how the films interact with water seems to be contradictory for different measurements (density compared to moisture content, for instance).
  • Water solubility, water adsorption, and swelling power have been revised accordingly in order for the characterization of the film concurrence with density, thickness and moisture content of the films.

  1. Other notes - Text says that Figure 3a is ES/CS composite, but the figure seems to show ES only.  Would be helpful to have all three (ES, CS, and ES/CS) for the FTIR.
  • The figures has been updated accordingly

  1. Error bars on the Storage Stability graph would be helpful.  Additionally, it doesn't look like there is data for ESP/CS film at room temp.
  • Error bars of the storage stability data have been updated. The legend for ESP/CS film has been changed to a more obvious colour and design.

Reviewer 4 Report

Thank you very much for choosing me as a reviewer of the manuscript entitled “

 Physicochemical Characterization of Biodegradable Eggshell Powder/Corn Starch Film Synergetic Application for Adsorption of Heavy Metal

The present paper has been carefully studied several times and some criticisms have been made that should be corrected.

1- Title

  • I recommended to change title and use shorter title.

2- Abstract

  • heavy metals; determine the heavy metals type in the abstract.
  • You should define SEM and FTIR
  • Add some more results in the abstract.
  • Keywords are not suitable, use relevant keywords.

3- Introduction

Literature survey is poor, In the introduction section, in relate edible films, biodegradable films, starch film, adsorption, heavy metals removing and so, there are so related references, so you should use the related following references:

  • A review of the applications of bioproteins in the preparation of biodegradable films and polymers. Journal of Chemistry Letters. 2020 Jul 1;1(2):47-58.
  • Biodegradable nanocomposite film based on gluten/silica/calcium chloride: physicochemical properties and bioactive compounds extraction capacity. Journal of Polymers and the Environment. 2021 Aug;29(8):2557-71.
  • Biodegradable nano composite film based on modified starch-albumin/MgO; antibacterial, antioxidant and structural properties. Polymer Testing. 2021 May 1;97:107182.
  • Design and fabrication of nanocomposite-based polyurethane filter for‏‏ improving municipal waste water quality and removing organic pollutants. Adsorption Science & Technology. 2019 Mar;37(1-2):95-112.

4- Material and methods

  • You should report all details of materials (purity and ….)
  • 2. Preparation of eggshell powder (ESP), needs to be rewrote
  • Use number the used formula in the manuscript.
  • Statistics analysis section is unclear, you should explain in basically.

5- Result and discussion

  • You should report full name of all abbreviations.
  • You used SEM figures with different magnifications, so the comparing of them is difficult.
  • In Table 1, determine the significance of the differences between the two types of films using lowercase letters
  • In the FTIR, please confirm the following claims with suitable references:
  • An intense peak of the ES particle is observed at 1400.03 cm−1 and a strong 312 absorption band, which can be strongly associated with the presence of carbonate 313 minerals within the ES matrix.
  • All Figures quality is so low, enhance them.

  1. Conclusion
  • The conclusion is short and poor, revise it.

7- Finally, it should be mentioned that there are so many scientific and grammatical errors in the text, it is suggested to revise the manuscript text carefully.  

Author Response

Reviewer 4

Thank you very much for choosing me as a reviewer of the manuscript entitled “Physicochemical Characterization of Biodegradable Eggshell Powder/Corn Starch Film Synergetic Application for Adsorption of Heavy Metal”. The present paper has been carefully studied several times and some criticisms have been made that should be corrected.

  1. Title: I recommend changing the title and using a shorter title.
  • Thank you for the recommendation. The title has been changed to “Development and characterization of the biodegradable film derived from eggshell and cornstarch”.

  1. Abstract:

heavy metals; determine the heavy metals type in the abstract. You should define SEM and FTIR. Add some more results in the abstract. Keywords are not suitable, use relevant keywords.

  • The abstract and keywords have been revised and updated according to the comments.

  1. Introduction:

Literature survey is poor, In the introduction section, in relate edible films, biodegradable films, starch film, adsorption, heavy metals removing and so, there are so related references, so you should use the related following references:

A review of the applications of bioproteins in the preparation of biodegradable films and polymers. Journal of Chemistry Letters. 2020 Jul 1;1(2):47-58.

Biodegradable nanocomposite film based on gluten/silica/calcium chloride: physicochemical properties and bioactive compounds extraction capacity. Journal of Polymers and the Environment. 2021 Aug;29(8):2557-71.

Biodegradable nanocomposite film based on modified starch-albumin/MgO; antibacterial, antioxidant and structural properties. Polymer Testing. 2021 May 1;97:107182.

Design and fabrication of nanocomposite-based polyurethane filter for improving municipal waste water quality and removing organic pollutants. Adsorption Science & Technology. 2019 Mar;37(1-2):95-112.

  • Thank you for the constructive comment. The introduction section has been revised accordingly which can be found in manuscript section 1. The authors have included the above-mentioned references as numbers [7], [8], [9] and [24].

  1. Material and methods

  1. i) You should report all details of materials (purity and ….)
  • All details of the materials have been reported in section 2.1.

  1. ii) Preparation of eggshell powder (ESP), needs to be rewrite
  • The preparation method of eggshell powder has been rewrote in section 2.2.

iii) Number the used formula in the manuscript.

  • All the formulas in the manuscript have been labeled with numbers [(1), (2), (3), (4), (5) and (6)].

  1. iv) Statistics analysis section is unclear, you should explain it basically.
  • Section 2.8 Statistics analysis has been revised and explained.

  1. Result and discussion:

  1. i) You should report the full name of all abbreviations.
  • The full name of all abbreviations has been added at the beginning of the manuscript.

  1. ii) You used SEM figures with different magnifications, so comparing them is difficult.
  • The SEM has been updated according to the comments

iii) In Table 1, determine the significance of the differences between the two types of films using lowercase letters

  • The statistics in the Table have been updated

  1. iv) In the FTIR, please confirm the following claims with suitable references:

An intense peak of the ES particle is observed at 1400.03 cm−1 and a strong 312 absorption band, which can be strongly associated with the presence of carbonate minerals within the ES matrix.

  • The references have been included as number [x].

  1. v) All Figures quality is so low, enhance them.
  • All the figures have been updated

  1. Conclusion

The conclusion is short and poor, revise it.

  • The conclusion has been revised and updated based on the suggestions.

  1. Finally, it should be mentioned that there are so many scientific and grammatical errors in the text, it is suggested to revise the manuscript text carefully.
  • Thank you for the comment. The manuscript has been revised carefully, where scientific and grammatical errors have been taken into account.

Reviewer 5 Report

Comments and Suggestions for Authors

Dear Authors,

The Title:

 Physicochemical Characterization of Biodegradable Eggshell Powder/Corn Starch Film Synergetic Application for Adsorption of Heavy Metal

I have to read your manuscript with great attention and interest.

The authors investigated a mixture of corn starch and shell powder. They made the mixture using the casting process. The blend is to be used as edible films with a high interest in removing heavy metals from food products.

The submission falls within the scope of the journal and is sufficiently original. I recommended the publication after MAJOR REVISIONS or REJECT.

  • Add a list of symbols in alphabetical order at the beginning of the manuscript
  • Introduction: demonstrate more strongly the practical use of the mixture in the food industry, through the support of the examples.
  • Change the settings in Fig. 2, put the legend next to the bar charts
  • The subject of the manuscript is completely divergent from the research and conclusions. Which heavy metals were taken into account, how was their adsorption with the blend checked?
  • Conclusions: Please write down your requests as points

Author Response

Reviewer 5

I have to read your manuscript with great attention and interest. The authors investigated a mixture of cornstarch and shell powder. They made the mixture using the casting process. The blend is to be used as edible films with a high interest in removing heavy metals from food products. The submission falls within the scope of the journal and is sufficiently original. 

  1. Add a list of symbols in alphabetical order at the beginning of the manuscript
  • The list of symbols have been added in alphabetical order at the beginning of the manuscript. 

  1. Introduction: demonstrate more strongly the practical use of the mixture in the food industry, through the support of the examples.
  • Thank you for the comment. The revisions have been provided in section 1 Introduction, paragraph 3. 

  1. Change the settings in Fig. 2, put the legend next to the bar charts
  • The legend was adjusted and placed next to the bar charts as shown in Fig. 2. 

  1. The subject of the manuscript is completely divergent from the research and conclusions. Which heavy metals were taken into account, how was their adsorption with the blend checked?
  • The manuscript is focusing on the development and characterization of the eggshell powder/cornstarch film which could be applied in the adsorption of heavy metals. Thus, the title has been revised into “development and characterization of the biodegradable film derived from eggshell and cornstarch” 

  1. Conclusions: Please write down your requests as points
  • The conclusion has been updated accordingly.

Round 2

Reviewer 2 Report

it can be published at this status

Reviewer 3 Report

The comments and concerns raised in the original review have been adequately addressed.  The revised manuscript is recommended for publication.

Reviewer 4 Report

It can be accepted.

Reviewer 5 Report

Thank you for responding to the comments.